# Assessing the Nonlinear Changes in Global Navigation Satellite System Vertical Time Series with Environmental Loading in Mainland China

Jie Zhang [1], Zhicai Li [1,*], Peng Zhang [2], Fei Yang [1], Junli Wu [2], Xuchun Liu [3], Xiaoqing Wang [2] and Qianchi Tan [1]

1   College of Geoscience and Surveying Engineering, China University of Mining and Technology-Beijing, Beijing 100083, China; sqt2100204092@student.cumtb.edu.cn (J.Z.); yangfei@cumtb.edu.cn (F.Y.); 2110260107@student.cumtb.edu.cn (Q.T.)
2   Department of Geodesy, National Geomatics Center of China, Beijing 100830, China; zhangpeng@ngcc.cn (P.Z.); jlwu@ngcc.cn (J.W.); xqwang@ngcc.cn (X.W.)
3   Shool of Geomatics and Urban Spatial Informatics, Beijing University of Civil Engineering and Architechture, Beijing 102616, China; liuxuchun@bucea.edu.cn
*   Correspondence: zcli@cumtb.edu.cn

**Abstract:** This study investigated the nonlinear changes in the vertical motion of 411 GNSS reference stations situated in mainland China and assessed the influence of the environmental load on their vertical displacement. The researchers evaluated the effect of environmental load by calculating the change in annual cycle amplitude before and after its removal, focusing on its impact across regions with distinct foundation types. The results demonstrate that removing the environmental load led to a considerable reduction of approximately 50.25% in the annual cycle amplitude of vertical motion for GNSS reference stations in mainland China. This reduction in amplitude improved the positioning accuracy of the stations, with the highest WRMS reduction being 2.72 mm and an average reduction of 1.03 mm. The most significant impact was observed in the southwestern, northern, and northwestern regions, where the amplitude experienced a notable decrease. Conversely, the southeastern region exhibited a corresponding increase in amplitude. This article innovatively explored the effects of environmental loads on diverse foundation types. When categorizing GNSS reference stations based on their foundation type, namely, bedrock, 18 m soil layer, and 4–8 m soil layer stations, this study found that removing the environmental load resulted in reductions in annual cycle amplitudes of 49.37%, 59.61%, and 46.48%, respectively. These findings indicate that 18 m soil layer stations were more susceptible to environmental load-induced vertical motion. In conclusion, the impact of the environmental load was crucial when analyzing the vertical motion of GNSS reference stations in mainland China, as it was essential for establishing a high-precision coordinate reference framework and studying the tectonic structure of the region.

**Keywords:** global navigation satellite systems; environmental loading; nonlinear motion; time series



## 1. Introduction

The advancement in global navigation satellite system (GNSS) technology has led to a substantial improvement in its accuracy, providing high spatial and temporal resolution with extensive coverage. As a result, GNSS measurements have gained increasing significance in the fields of geodesy and geodynamics [1–3]. Long-term GNSS time series analysis has been instrumental in revealing both linear and nonlinear changes in the behavior of measurement stations [2,4–8]. The nonlinear variations are widely acknowledged to be influenced by various geophysical factors, including atmospheric tides, oceanic tides, and hydrological loads that impact the GNSS stations [9–17].

The impact of atmospheric loading models on GNSS stations exhibits variation, but their removal from inland stations has shown a reduction in the vertical weighted root-mean-square (WRMS) error by approximately 3–4% [18]. The effectiveness of atmospheric

loading correction is more pronounced for stations located near the equator in correcting their vertical displacement; however, it may lead to increased errors in other regions. To model non-tidal atmospheric loading, a daily average correction is applied to daily coordinate estimates, resulting in a greater improvement in the vertical root mean square (RMS) when implemented at the observation level [19,20].

The Influence of oceanic non-tidal loading on GNSS stations is intricate. Ocean tides were found to reduce the vertical WRMS of coastal GNSS stations by around 10% [11], and in some cases, the reduction can reach up to 20–30% for specific stations [12,21]. Hydrological loading also affects GNSS stations and contributes to variations in the GNSS time series, with its phase preceding the GNSS time series phase [22,23]. The research outcomes unveil a distinctive interaction between hydrological loading and thermal expansion phenomena. This implies that certain displacements induced by hydrological loading could potentially counteract those stemming from thermal expansion influences, as corroborated by a recent investigation [24]. It is crucial to emphasize this specific observation in relevant scholarly inquiries. Despite the ability of GRACE (Gravity Recovery and Climate Experiment) data to explain a larger portion of the GPS vertical annual signal compared with environmental load models, its limited temporal resolution and accuracy restrict its capability to remove environmental loading from the GNSS daily time series [25]. Environmental loads exhibit spatial variability based on geographic factors, such as altitude and latitude for atmospheric loads; distance from the ocean for oceanic non-tidal effects; and local factors like precipitation, vegetation cover, and humidity for hydrology [16,20,26,27]. Additionally, the use of different environmental load models by various agencies can lead to discrepancies between the models [28].

Scholars have extensively investigated the impact of various environmental loads, including tidal, atmospheric, snow and soil water, and oceanic non-tidal loads, on GNSS stations. Several studies provided valuable insights into the reduction in vertical seasonal variation amplitudes by removing these loads. For instance, a study that focused on 25 Chinese crustal motion observation network reference stations revealed that the removal of environmental loads led to a significant reduction of approximately 37% in the vertical seasonal variation amplitude [29]. Similarly, a study conducted in Southern California involving 22 IGS stations demonstrated that removing environmental loads (such as atmospheric pressure, soil moisture, snow depth, and non-tidal ocean) resulted in a reduction in the seasonal amplitude of the vertical time series by approximately 12.70–21.78% despite variations in the effects of environmental loads between adjacent geographic regions [30]. In mainland China, the removal of environmental loads (including atmospheric load, land storage, and non-tidal ocean) from 224 GPS vertical time series that had been continuously operating for over 4 years led to a reduction of about 34% in the weighted root-mean-square value [31]. The southwest region exhibited significant seasonal variability, possibly due to the large uncertainty in its land storage. Moreover, environmental loads (atmospheric, hydrographic, and non-tidal ocean) were found to explain approximately 43% of the vertical deformation in the second reprocessed data from 900 global IGS stations [32]. Collilieuxa et al. reported that removing environmental loads led to a considerable reduction of 73% in the annual amplitude of GPS station time series in the ITRF framework [33,34]. Furthermore, Jiang et al. concluded that removing environmental loads contributed to mitigating around 70% of the nonlinear variation observed in GPS reference stations in mainland China [35]. These findings collectively underscore the crucial role of accounting for and removing environmental loads when analyzing GNSS station data, as it significantly impacts the accuracy and interpretation of vertical motion and deformation in various geographic regions.

Previous studies in mainland China were limited by a relatively small number of GNSS reference stations and a reliance on global IGS data, leading to challenges in fully comprehending the impact of environmental loads on these stations in the region. To address these limitations, this study focused on mainland China and utilized a compre-

hensive dataset of 411 national reference stations in the area. The primary objective of this study was to examine the influence of environmental loads on the vertical nonlinear variation of GNSS reference stations and to explore how these loads affect different types of GNSS reference stations. The findings of this research carry significant implications for advancing the understanding of climate-related mass redistribution and provide crucial insights for improving the reliability and precision of GNSS-based geodetic analyses in mainland China.

## 2. Data

### 2.1. Environmental Load Data

In this study, the data for hydrological loading (HYDL), non-tidal atmospheric loading (NTAL), and non-tidal ocean loading (NTOL) were obtained from the German Geosciences Center (GFZ). The data had a spatial resolution of $0.5° \times 0.5°$ and a temporal resolution of 3 h for NTAL and NTOL, while HYDL had a temporal resolution of 24 h. To calculate the environmental loading displacement data, the center-of-figure (CF) method was employed, which is similar to the center-of-mass (CE) method. For more detailed information, readers are encouraged to refer to the references [36–38].

### 2.2. GNSS Data

The GNSS data utilized in this study originated from two major projects: the Crustal Movement Observation Network of China (CMONOC) and the China Modern Geodetic Datum Infrastructure Construction (CMGDIC). The CMONOC Project, which spanned from 2011 to 2020, established 260 national GNSS reference stations, while the CMGDIC project, carried out from 2015 to 2020, involved the construction of 210 national GNSS reference stations [39–41].

The processing of raw data was conducted using BERNESE 5.2 software [42], encompassing daily solution execution and requisite adjustments. The data processing within the scope of daily solutions encompassed stages such as data preprocessing, baseline combination, integer ambiguity resolution, and independent network solution computation. For the resolution of integer ambiguities, a strategy incorporating both pseudorange and phase processing modes was employed. This involved the utilization of a quasi-ionosphere-free (QIF) wide-lane (WL) ambiguity resolution approach based on the pseudorange, alongside an L5 strategy grounded in phase data.

To ensure a consistent regional context across China and establish a linkage with the global framework during daily solutions, a collaborative approach was employed. This involved six strategically positioned International Global Navigation Satellite System (GNSS) Service (IGS) stations located throughout China and its adjacent regions, namely, BJFS, DAEJ, IISC, POL2, TCMS, and ULAB. During the process of daily network adjustment, solution-independent exchange (SINEX) files sourced from global daily solutions provided by the CODE Data Processing and Analysis Center were obtained. These files were subsequently converted into normal equation files and integrated with the existing normal equation files of daily solutions. This integration was achieved by anchoring the aforementioned six IGS stations and global framework stations, resulting in the amalgamation of regional daily solutions with global framework daily solutions.

In line with baseline constraints, all stations operating within the IGB08 framework were employed as baseline framework points. Furthermore, a uniform a priori unit weight mean error of 0.001 m was assigned to all reference points, thereby ensuring the integrity of the baseline constraints [43–45]. In the context of the least-squares constraint, priority was given to the adoption of the global seven-parameter transformation method over the block adjustment three-parameter transformation.

In this study, a rigorous selection process was employed to choose 411 out of the initial 470 national reference stations. Stations with over 50% missing data, time series shorter than three years, excessive errors, and those located outside of China were excluded from the analysis. The distribution of the selected stations is visually presented in Figure 1,

which illustrates their spatial arrangement within mainland China. Green and grey circles refer to the sites used and excluded, respectively, in the study.

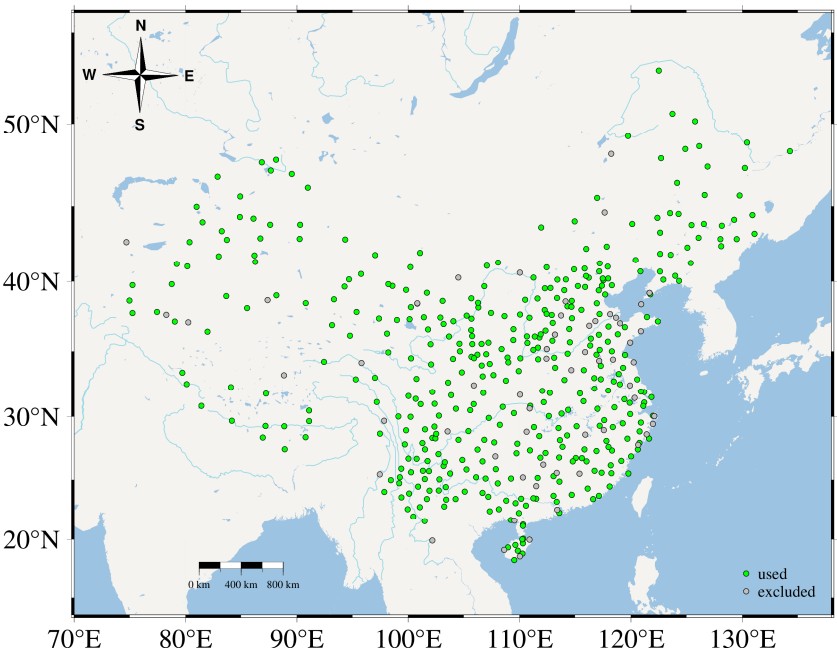

**Figure 1.** Spatial dispersion of accessible GNSS stations.

## 3. Data Processing

In the context of this research, data amalgamation from GFZ led to the determination of aggregate vertical displacements, denoted as SLMs (surface loading models). Together, the SLMs comprehensively encapsulated the cumulative elevation alterations arising from NTAL, NTOL, and HYDL. To correct the GNSS coordinate time series, the SLMs were subtracted from it. The corrected GNSS coordinate time series accounted for the influence of environmental loads on the vertical displacement of the GNSS reference stations. To further refine the data, both the original time series and the corrected time series were processed using Hector 1.6 software using the following steps, including the removal of coarse differences; fitting trend terms; and incorporating periodic terms, such as annual and semi-annual variations [46]. Subsequently, the amplitude and phase of the time series were calculated, providing a more detailed and accurate representation of the vertical motion of the GNSS reference stations.

To assess the influence of the total environmental load displacement on the seasonal variation in GNSS vertical displacement, this research employed two metrics as evaluation measures, i.e., the weighted average amplitude reduction rate and the weighted root-mean-square error difference. These metrics were utilized to quantitatively analyze and understand the impact of the SLMs on the seasonal changes observed in the GNSS vertical displacement data.

The weighted average amplitude reduction rate serves as the first metric used to quantify which SLMs contributed to the reduction in seasonal variation observed in the GNSS vertical time series [32]. By calculating the weighted average amplitude reduction rate, the researchers can gain valuable insights into the influence of the SLMs on the seasonal changes in the GNSS vertical displacement data, providing a comprehensive understanding of the role of environmental loads in shaping the observed variations.

$$ratio(SLMs) = \frac{wtmaGNSS - wtma(GNSS - SLMs)}{wtma(GNSS)} \times 100\% \qquad (1)$$

where $wtma(GNSS)$ denotes the weighted average amplitude of the GNSS reference station and $wtma(GNSS - SLM)$ denotes the weighted average amplitude of GNSS after removing the SLMs. Take $wtma(GNSS)$ for example, which is calculated using the following formula:

$$wtma(GNSS) = \frac{\sum_{i=1}^{n} w_{GNSS(i)} \times AM_{GNSS(i)}}{\sum_{i=1}^{n} w_{GNSS(i)}} \tag{2}$$

where $w_{GNSS(i)}$ denotes the weight of the $i$th GNSS reference station and the formal error sigma, given by the Hector 1.6 software, is

$$w_{GNSS(i)} = \frac{1}{sigma(i)^2} \tag{3}$$

where $AM_{GNSS(i)}$ denotes the amplitude of the $i$th GNSS reference station.

The weighted root-mean-square error difference is determined by calculating the weighted root-mean-square error (WRMS) of the GNSS time series before and after the removal of the SLMs. The difference between these two WRMS values reflects the magnitude of the accuracy improvement in the GNSS time series [2]. By assessing the WRMS error difference, the researchers can gauge the extent to which the removal of the SLMs enhances the accuracy of the GNSS vertical time series. This provides valuable information regarding the significance of environmental load corrections in refining and improving the precision of GNSS measurements in the context of vertical displacement analysis.

$$diff = WRMS_{Before} - WRMS_{After} \tag{4}$$

## 4. Data Analysis and Discussion

### 4.1. Adaptation of the SLMs in Different Areas of Mainland China

The initial analysis in this study focused on investigating the impact of the SLMs on different areas of mainland China using the amplitudes and phases obtained from fitting the SLMs and GNSS time series. Former studies emphasized the vertical time series, as environmental loads have less effect on the horizontal direction of GNSS reference stations, and the annual variation of the GNSS vertical time series exhibits a larger amplitude compared with the semi-annual variation [29,32,47,48]. Figure 2a,b present the annual variations in the SLMs and GNSS time series for the 411 stations, respectively. In Figure 2, the different colors indicate the size of the annual amplitude, with purple representing amplitudes larger than 10 mm. The arrow direction illustrates the initial phase and the clockwise rotation angle with respect to the north direction indicates the phase size, both of which were obtained by fitting a sine function to the time series. The results indicate that the phases of the SLMs and GNSS time series were largely similar, except for a phase difference observed in the southeast region. Moreover, the largest amplitudes were found in the southwestern region, followed by the north and northwest of mainland China. In contrast, amplitudes in other regions were mostly less than 5 mm. This suggests a good spatial consistency between the SLMs and GNSS vertical time series.

Figure 3a–d display the annual amplitude and phase of GNSS, NTAL, NTOL, and HYDL, respectively. It is evident that NTAL exhibited better agreement with GNSS compared with the other two environmental loads. There was a significant difference in the magnitude of NTAL amplitude between North and Central China and other regions. This variation may be attributed to the altitude influence, where lower altitudes in North and Central China experience a larger atmospheric load, while lower latitudes in South China lead to a larger atmospheric load due to the lower altitude [26]. The amplitude of NTOL was found to be greater at stations situated along the coastline than those farther from the coastline. Additionally, the amplitude of HYDL in mainland China was mainly the largest in the southwestern region, which was likely due to the substantial hydrological load generated by the annual monsoon in South and Southeast Asia [22,49]. Moreover, the annual amplitudes of SLMs and GNSS in North and Southwest China significantly

differ from those in other regions. This discrepancy was attributed to North and Central China being strongly influenced by atmospheric loads, while Southwest China is strongly impacted by hydrological loads. Overall, these findings highlight the spatial variations in the amplitudes and phases of environmental loads across different regions in mainland China, as well as their agreement with GNSS data, providing valuable insights into the specific influences of different environmental loads on the GNSS vertical displacement in the region.

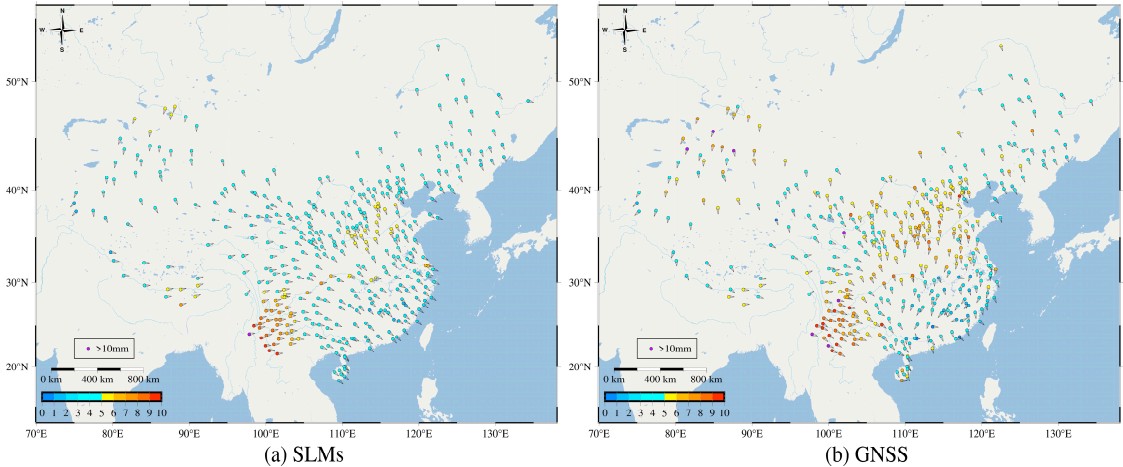

**Figure 2.** (**a**) Annual amplitude and phase acquired through fitting SLMs time series. (**b**) Annual amplitude and phase acquired through fitting GNSS time series.

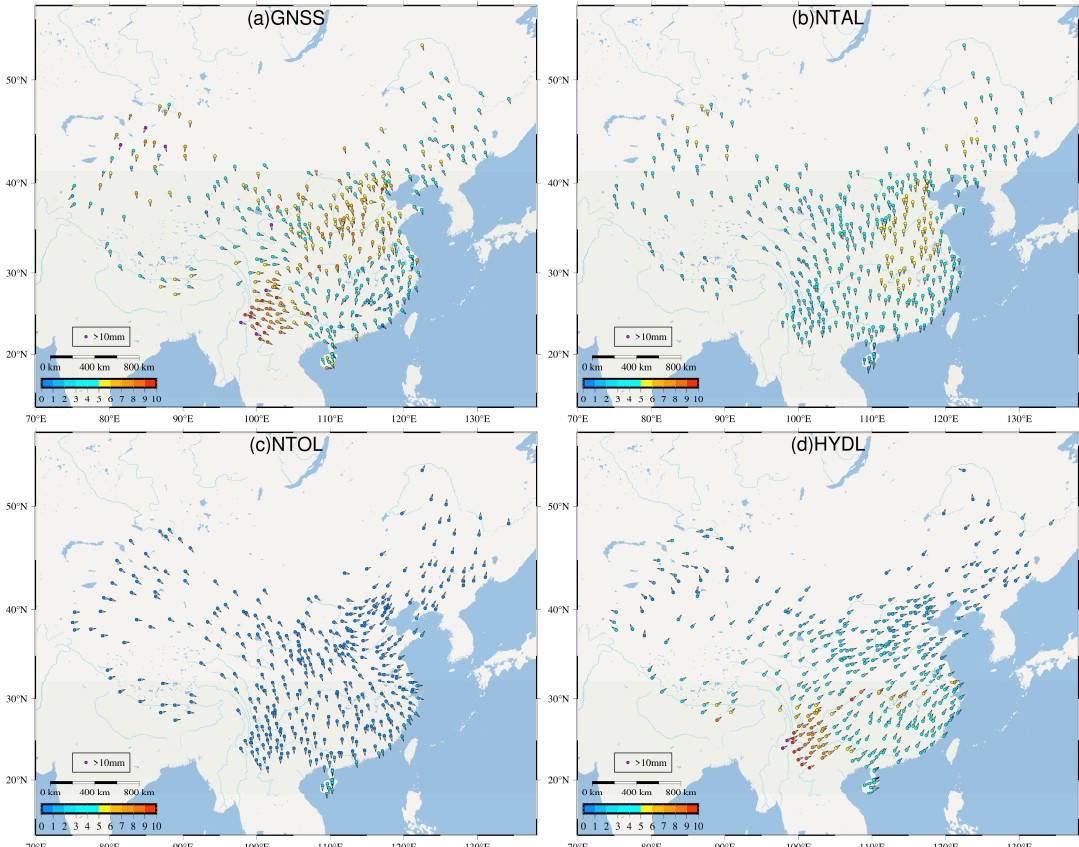

**Figure 3.** (**a**) Annual amplitude and phase derived from fitting time series data of the GNSS. (**b**) Annual amplitude and phase obtained by fitting time series data of NTAL. (**c**) Annual amplitude and phase acquired by fitting time series data of NTOL. (**d**) Annual amplitude and phase derived from fitting time series data of the HYDL.

### 4.2. Effect of SLMs on the Vertical Displacement of GNSS in the Mainland of China

The data presented in Table 1 are the statistics of annual amplitude changes before and after the SLMs removal, which demonstrates that the removal of the SLMs from the GNSS vertical time series had a significant impact on reducing the annual amplitude of the GNSS signal. Specifically, the annual weighted average amplitude of the original GNSS time series was 5.15 mm, whereas the annual weighted average amplitude of the GNSS time series after removing the SLMs (GNSS-SLMs) was 2.78 mm. This reduction in the annual amplitude after eliminating SLMs indicates that environmental loads contributed to 50.25% of the GNSS vertical nonlinear variation.

**Table 1.** Alterations in annual amplitude before and after SLMs exclusion (unit: mm).

|  | Weighted Average Amplitude | Maximum Amplitude | Minimum Amplitude |
| --- | --- | --- | --- |
| GNSS | 5.13 | 12.40 | 0.87 |
| GNSS-SLMs | 2.77 | 10.87 | 0.42 |

This indicates that the impact of the SLMs on the vertical direction of GNSS reference stations in mainland China was greater compared with findings from previous studies. For instance, Wang et al. reported a 37% reduction in the amplitude of vertical seasonal variation after removing SLMs from the data collected at 25 Chinese crustal motion observation network benchmark stations [29]; Niu et al. found that SLMs can explain 42.99% of the vertical seasonal variation of global IGS stations and 58.69% of the vertical seasonal variation in the European region [32].

In contrast, the present study's analysis of 411 GNSS reference stations in Mainland China demonstrates a more significant influence of the SLMs on the vertical time series, resulting in a greater reduction in the amplitude of the vertical seasonal variation. The disparity could be attributed to differences in the geographic locations, station density, and environmental loading characteristics between the various studies.

Figure 4 presents a histogram illustrating the distribution of amplitude changes at GNSS reference stations following the removal of the influence of SLMs. It reveals that 89.5% of the stations experienced a reduction in amplitude after the removal of SLMs. Specifically, a significant proportion of stations exhibited amplitude reductions in the range of 2–4 mm. This observation indicates that the removal of SLMs had a beneficial impact on the GNSS time series, effectively reducing the annual amplitude. The findings suggest that the corrections for environmental loads, such as NTAL, NTOL, and HYDL, played a crucial role in enhancing the accuracy and precision of the GNSS observations. By removing the influence of the SLMs, this study demonstrated that the seasonal variations in the GNSS vertical displacement data could be better understood and characterized. He et al. achieved analogous outcomes through the utilization of GNSS data sourced from 206 IGS stations [50].

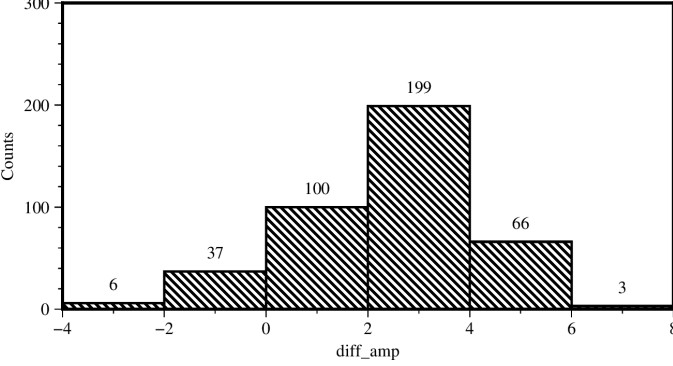

**Figure 4.** Histogram of annual amplitude change before and after the removal of SLMs.

Figure 5 provides a visual representation of the distribution of GNSS amplitude change after removing the influence of the SLMs. The plot shows that the majority of the stations indicated by green points exhibit a decrease in amplitude, with a notable concentration in regions of southwestern China, northern China, and the northwestern frontier areas. Conversely, there were relatively fewer stations depicted by red points displaying an increase in amplitude, which were primarily located along the Yangtze River in southeastern and southern China. The size of the points in the plot corresponds to the magnitude of the amplitude change, with larger points indicating a more substantial change in amplitude. This visual representation enables a clear understanding of the spatial distribution of amplitude changes across different regions in mainland China after the removal of the SLMs.

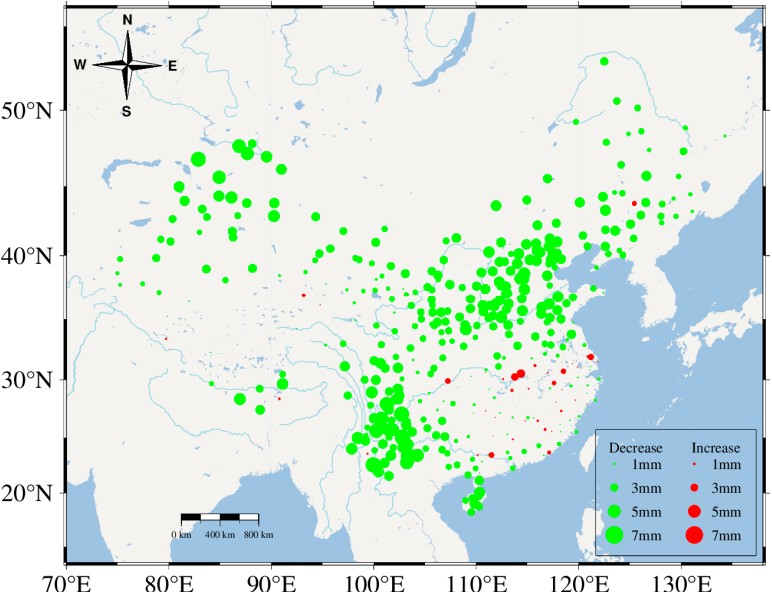

**Figure 5.** Spatial distribution of annual amplitude changes before and after the removal of SLMs.

The regions with notable amplitude reductions in the GNSS data were primarily situated in the southwestern region, northern China, and the northwestern border areas. In the southwestern region, the amplitude reduction varied, being more significant in the eastern part and relatively smaller in the western part. This pattern could be attributed to the presence of the Jinsha River–Red River rift zone, which divides the province of Yunnan into eastern and western regions. These two sub-regions exhibit substantial differences in terms of precipitation and surface relief [47].

By comparing Figure 2a (annual variations in the SLMs) and Figure 2b (annual variations in the GNSS time series), interesting observations can be made regarding the initial phase direction in different areas of mainland China. Specifically, in southeastern and southern China, the initial phase direction of the SLMs appeared to be more regular and consistent compared with the GNSS time series. On the other hand, the GNSS time series in the same region show a more chaotic and disordered pattern in terms of the initial phase direction. This discrepancy suggests the presence of multiple mass loads with different displacement directions in southeastern and southern China. These various mass loads may lead to a complex interplay of forces, resulting in a compensation effect that appears as a disordered regional load [32]. The regularity observed in the initial phase directions of the SLMs may be indicative of the dominant influence of specific environmental loads in the region, which can help in understanding the underlying geodetic processes and the impact of different environmental loads on the GNSS vertical displacement in this area.

The analysis of GNSS improvement after removing the SLMs revealed a correlation with the distribution of the SLMs amplitude presented in Figure 2a. Specifically, regions in northern China and southwestern China, where the SLMs amplitude was higher, show

larger improvements in the GNSS measurements after the removal of environmental loads. This alignment between the distribution of the SLMs amplitude highlights the magnitude of the significance of correcting for SLMs to achieve accurate GNSS measurements. Nevertheless, one notable exception was observed: along the Yangtze River, the original amplitude was higher, but the amplitude increased.

To assess the impact of removing the SLMs effect on the accuracy of the GNSS vertical time series, this study computed WRMS values before and after the SLMs removal and calculated the difference. Table 2 presents the percentage of change in the WRMS values after the SLMs removal for each station. It can be seen that 391 stations accounted for 95.1% of the total number of stations that experienced a reduction in WRMS values, indicating an improvement in the positioning accuracy of these stations following the removal of the SLMs effect. Among these stations, 55% of the stations demonstrate a reduction in WRMS values by 0–1 mm, 34.1% of the stations exhibit a reduction of 1–2 mm, and 6% of the stations show a reduction of 2–2.719 mm. These findings aligned with the results reported by Jiang et al. [2], confirming the positive impact of SLMs removal on the accuracy of the GNSS vertical time series. However, they differ from the results obtained by Gu et al. [29], which may be attributed to differences in the WRMS calculation method. This study employed a WRMS calculation method consistent with the method in [2], while the specific method for WRMS calculation used by Gu et al. was not disclosed in their paper. The consistency in the results between this study and Jiang et al.'s research further validated the effectiveness of removing the SLMs effect in enhancing the accuracy of GNSS measurements, providing crucial insights for improving the reliability and precision of GNSS-based geodetic analyses in mainland China.

**Table 2.** Improvement of WRMS after SLMs removal.

| Range of Improvement (mm) | −1.25–0 | 0–1 | 1–2 | 2–2.719 |
|---|---|---|---|---|
| Percentage (%) | 4.86 | 54.98 | 34.06 | 6.08 |

Figure 6 illustrates the spatial distribution of changes in the WRMS of the GNSS reference station sequences following the removal of the SLMs effect. The green points in the figure indicate a decrease in the WRMS of the station sequences, which corresponds to an increase in accuracy. The red points represent an increase in WRMS, signifying a decrease in accuracy after the SLMs removal. The sizes of the points reflect the magnitude of the change in the WRMS. This reveals that the reduction in WRMS was relatively minimal at GNSS reference stations situated in the Tibetan Plateau, Sichuan Basin, and coastal regions, which implies that the removal of the SLMs effect had a limited impact on the accuracy of these stations in these areas. On the other hand, the reduction in the WRMS was more substantial at the GNSS reference stations in the northwestern region, North China, and Southwest China, indicating a significant improvement in the accuracy after the SLMs removal for stations in these regions. These distribution results are in alignment with the findings reported by Gu et al. [31], further validating the effectiveness of SLMs removal in enhancing the accuracy of GNSS measurements in mainland China.

Note that 43 stations did not exhibit a decrease in the annual amplitude after removing the SLMs effect. To investigate the reasons behind this phenomenon, a comparison was made between the GNSS and SLMs time series for each station. Two main reasons were identified for the lack of amplitude reduction after the SLMs removal: Within the dataset, there were 29 stations whose raw GNSS time series exhibited either weak annual periodicity or periodicity weaker than that of the SLMs. The SLMs, characterized by a robust annual periodicity, engendered a discernible influence. Consequently, the elimination of the SLMs yielded GNSS time series that exhibited enhanced annual periodic behavior compared with their initial states. Additionally, among the stations, 14 displayed substantial phase disparities between the initial phases of their GNSS time series and the SLMs. Specifically, at the peak of the GNSS time series waveform, SLMs corresponded to its nadir. This variance underscored that subsequent to the SLMs removal, the amplitude of the annual

cycle within the GNSS time series became magnified. These two reasons accounted for the lack of amplitude reduction in certain stations after the SLMs removal.

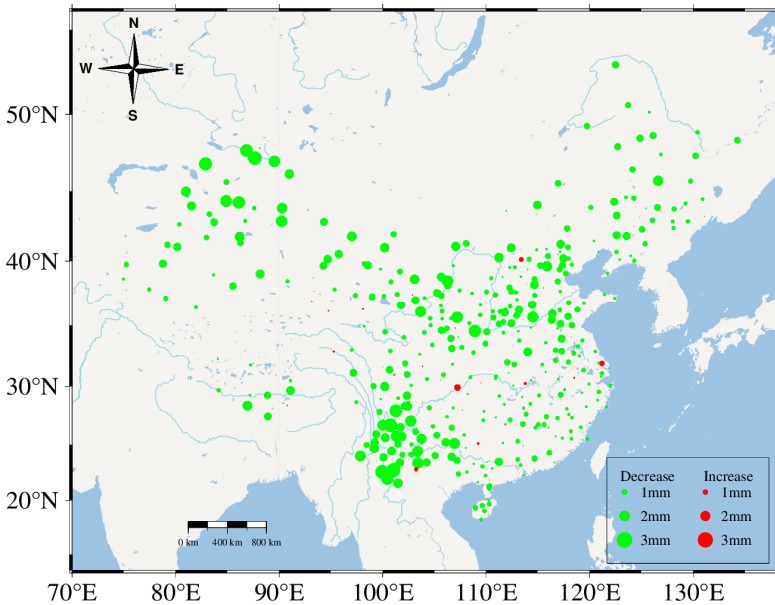

**Figure 6.** Spatial distribution of WRMS changes before and after the removal of SLMs.

Figure 7 illustrates the periodic behavior of raw GNSS time series from three representative GNSS reference stations, namely, BJSH, JSNT, and GDSG. Each figure displays different characteristics of the raw GNSS time series and its relationship with the SLMs time series. In Figure 7a, the raw GNSS time series of the BJSH station exhibits a clear amplitude and strong annual periodicity, with a phase similar to that of the SLMs time series. The amplitude of the green line, which represents the GNSS time series after removing the SLMs, is noticeably lower than the black line, indicating a significant reduction in the annual period amplitude after the SLMs removal. In Figure 7b, the raw GNSS time series of the JSNT station displays a lower amplitude compared with the SLMs time series, leading to an increase in the annual amplitude after removing the SLMs. The black line representing the raw GNSS time series shows a weaker periodicity than the yellow line, which represents the SLMs time series. In Figure 7c, the raw GNSS time series of the GDSG station exhibits a weak annual amplitude and periodicity. After removing the SLMs, the GNSS time series shows a stronger periodicity and increased amplitude. This behavior could be attributed to the fact that the noise signal at this station was much higher than the environmental load signal, resulting in a weaker response to the environmental load. These observations provide insights into the complex relationships between the raw GNSS time series and SLMs time series at different reference stations. The variations in amplitude and periodicity after the SLMs removal highlight the diverse responses of GNSS stations to environmental loads, emphasizing the importance of accurate environmental load corrections for reliable and precise GNSS measurements and geodetic analyses.

Figure 8 illustrates the increase in amplitude resulting from the significant discrepancy between the GNSS time series and SLMs phase using two selected GNSS reference stations as examples. Figure 8b,c represent the AHQM and QHGE stations, respectively. The observations in Figure 8 are consistent with those in Figure 7, highlighting the phase difference between the fluctuations of the GNSS time series and the SLMs time series. Specifically, for the AHQM station, the GNSS time series reached its peak while the SLMs time series was still far from reaching its peak. This lag between the GNSS and SLMs time series indicates that environmental loads tended to follow a delayed response compared with the GNSS time series, especially for the hydrological loads. A similar pattern was observed for the QHGE station, where the GNSS time series exhibits fluctuations ahead of the SLMs time series. The lag in environmental loads, especially hydrological loads,

relative to the GNSS time series was reported in previous research by Gu et al. [31], who found that hydrological loads typically lag behind GNSS by three to four months. For GNSS reference stations located near the Yangtze River, the amplitudes were relatively high, but they increase after removing the SLMs. This phenomenon could be attributed to the high hydrological loads in the area and the lag between the SLMs and GNSS, resulting in a significant phase difference. Consequently, the amplitudes of the GNSS reference stations near the Yangtze River increased after removing the SLMs.

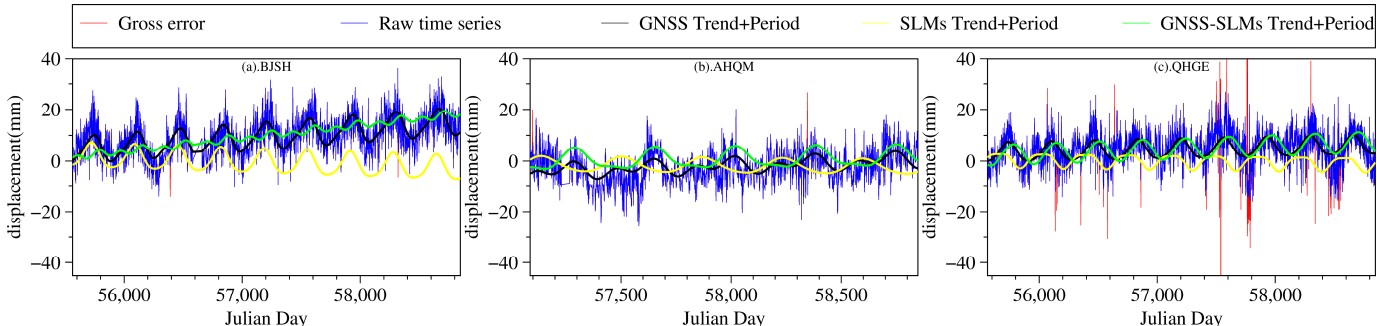

**Figure 7.** The fitted line (trend + period) was obtained from GNSS (black), SLMs (yellow), and GNSS after removal of the SLMs (green). (**a**) BJSH had a strong periodicity, (**b**) JSNT had a weaker GNSS period than SLMs, and (**c**) GDSG had a weak periodicity.

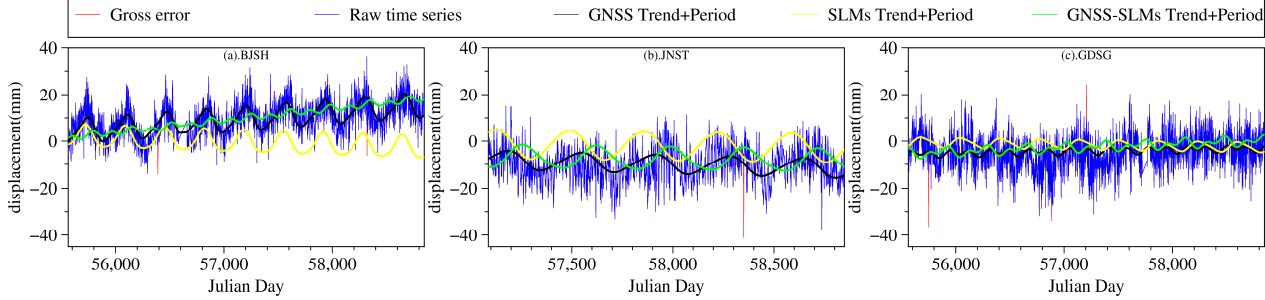

**Figure 8.** The fitted line (trend + period) was obtained from GNSS (black), SLMs (yellow), and GNSS after removal of SLMs (green). (**a**) GNSS and SLMs phase synchronization of BJSH, (**b**) GNSS and SLMs phase desynchronization of AHQM, and (**c**) QHGE and AHQM are the same.

### 4.3. Effect of SLMs on the Vertical Displacement of GNSS Reference Stations with Different Foundation Types

To understand the impact of the SLMs on the GNSS vertical non-linear changes, this study investigated the influence of the SLMs on the GNSS reference stations with different foundation types. In mainland China, the GNSS reference stations primarily fall into two categories, namely, rock-based stations and soil-based stations, which are further subdivided into 18 m soil stations and 4–8 m soil stations [39]. Figure 9 provides an illustration of the distribution of these different foundation types and their relation to fault zones. It is evident that the red triangles, representing 18 m soil stations, are predominantly situated near fault zones with east–west and north–south orientations. Additionally, there are no 18 m soil stations in the southeast and South China regions. In contrast, the blue squares, representing 4–8 m soil stations, are mainly distributed in eastern China. The green circles, indicating rock stations, are relatively evenly distributed throughout mainland China. The distinction in the distribution of different foundation types and their proximity to fault zones is crucial for understanding the differential impact of the SLMs on the GNSS measurements in various regions.

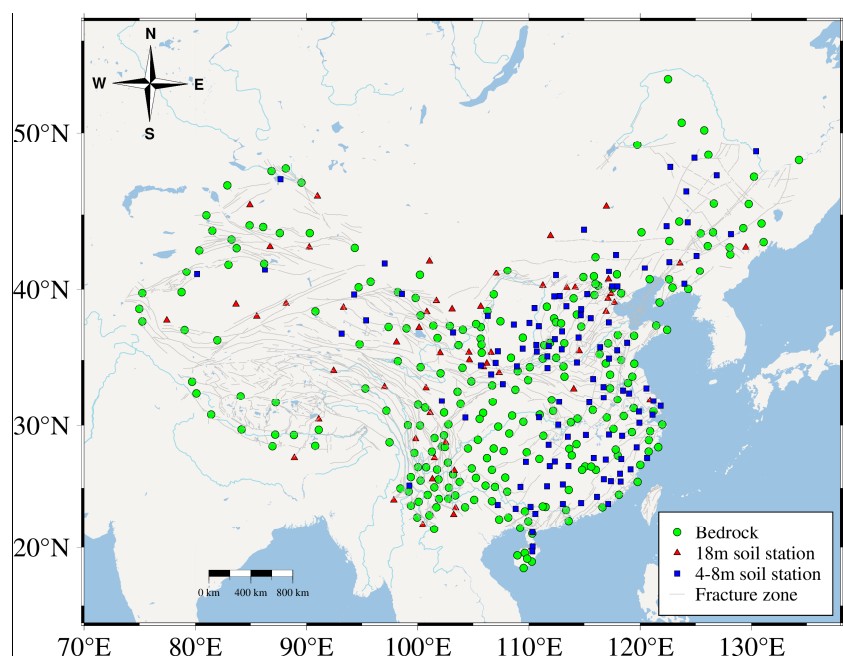

**Figure 9.** Distribution of GNSS reference stations and fracture zones for different foundation types.

Table 3 provides a summary of the weighted average amplitude change observed in different foundation types of GNSS reference stations. Among the total of 244 rock stations, 90.6% of them experienced a decrease in station amplitude after removing the SLMs. For the 53 18 m soil stations, only 1 station did not show a decrease in amplitude after the SLMs removal. Additionally, out of the 116 4–8 m soil stations, 83.6% of them exhibited a decrease in amplitude after removing the SLMs. These results indicate that the impact of the SLMs on the GNSS reference stations varied depending on their foundation type. Rock stations generally experienced a significant reduction in amplitude after the SLMs removal, with a high proportion showing a decrease. On the other hand, for 18 m soil stations, nearly all stations showed a decrease in amplitude after the SLMs removal, except for one station. The 4–8 m soil stations also exhibited a notable reduction in amplitude, with a majority showing a decrease.

**Table 3.** Weighted average amplitude of GNSS reference stations of different foundation types.

| Foundation Type | Number | Number of Reduction | Percentage (%) | GNSS_waa (mm) | G-S_waa (mm) | Waadr (%) |
|---|---|---|---|---|---|---|
| Bedrock | 244 | 221 | 90.57 | 4.95 | 2.51 | 49.37 |
| 18 m soil station | 55 | 54 | 98.18 | 5.44 | 2.19 | 59.61 |
| 4–8 m soil station | 112 | 93 | 83.03 | 4.82 | 2.40 | 46.48 |

Note: GNSS_waa is the GNSS weighted average amplitude, G-S_waa is the GNSS-SLMs weighted average amplitude, and Waadr is the weighted average amplitude decay rate.

In Table 3, the impact of removing the SLMs on the annual period amplitude is summarized for different types of GNSS reference stations. Among the 18 m soil stations, the largest change in annual period amplitude was observed, with an average decrease of 3.25 mm, representing 59.61% of the original amplitude. For the rock stations, the average decrease in annual period amplitude was 2.44 mm, accounting for 49.37% of the original amplitude. The 4–8 m soil stations exhibit the smallest change in annual period amplitude, with an average decrease of 2.42 mm, accounting for 46.48% of the original amplitude. These findings reveal that the 18 m soil stations were the most affected by the removal of the SLMs, as they experienced the largest reduction in annual period amplitude. Rock

stations also show a significant decrease in amplitude, while the 4–8 m soil stations display a comparatively smaller reduction.

Based on the data presented in Table 4, we can observe that the WRMS values of the average original GNSS time series were 7.95 mm, 10.18 mm, and 8.45 mm for rock stations, 18 m soil stations, and 4–8 m soil stations, respectively. It can be concluded that the stability of different foundation types of reference stations follows the order rock stations > 4–8 m soil stations > 18 m soil stations. This phenomenon may be attributed to the geographical distribution of these stations. As mentioned earlier, 18 m soil stations are primarily situated near fault zones, which might subject them to more complex geophysical influences and environmental loads, leading to higher variability and potentially reducing their stability compared with rock stations and 4–8 m soil stations.

**Table 4.** Average value of WRMS of GNSS reference stations of different foundation types (unit: mm).

| Foundation Type | WRMS (GNSS) | WRMS (GNSS-SLMs) | WRMS (GNSS)-WRMS (GNSS-SLMs) |
|---|---|---|---|
| Bedrock | 7.95 | 6.92 | 1.03 |
| 18 m soil station | 10.18 | 9.09 | 1.09 |
| 4–8 m soil station | 8.45 | 7.47 | 0.98 |

As demonstrated in Figure 6, the removal of the SLMs resulted in an increase in the GNSS time series amplitude in the Southeast and South China regions, contrary to the overall trend observed in other regions. Based on Table 3, we can deduce that the weighted average amplitude decline rate of the 18 m soil stations was the highest, and there were no 18 m soil stations in the Southeast and South China regions. Conversely, the weighted average amplitude decline rate of 4–8 m soil stations was the lowest, and approximately half of these stations were located in the Southeast and South China regions.

The choice of different foundation types for GNSS reference stations was primarily influenced by the local geological conditions, and the distribution of foundation types reflects the underlying geological variations. By considering the information in Table 4, we can infer that geological conditions near fault zones were often more unstable, rendering 18 m soil stations more susceptible to the influence of the SLMs. To validate this inference, we further examined the impact of the SLMs on the vertical non-linear movement of the GNSS reference stations in the North–South Fault Zone (95°E–105°E) (as presented in Table 5). This zone comprises 60 rock stations, 21 18 m soil stations, and 7 4–8 m soil stations. The maximum impact of the SLMs on 18 m soil stations was 58.07%, and on rock stations, this was 52.81%. The influence of the SLMs on rock stations near the North–South Fault Zone has been found to be higher, further supporting the notion that GNSS reference stations in geologically unstable areas are more susceptible to the influence of SLMs.

**Table 5.** Effect of SLMs in the vertical nonlinear motion of GNSS reference stations in the North–South Fault Zone region (95°E–105°E).

| Foundation Type | Weighted Average Amplitude Decay Rate | Number of Reference Stations | Number of Undecreased Stations |
|---|---|---|---|
| Bedrock | 52.81% | 60 | 1 |
| 18 m soil station | 58.07% | 21 | 0 |
| 4–8 m soil station | 45.29% | 7 | 0 |

Conversely, the Southeast and South China regions exhibit complex environmental conditions, with non-tidal atmospheric, non-tidal oceanic, and hydrological loads better reflecting the non-linear structural movement of the region. The unique geological and environmental characteristics of these regions may contribute to the observed increase in amplitude after removing the SLMs.

## 5. Conclusions

This study investigated the impact of environmental loads on 411 GNSS reference stations in mainland China, leading to several significant findings, given below.

Eliminating environmental loads resulted in a 50.25% reduction in the vertical annual period amplitude of the GNSS time series in mainland China. This reduction in amplitude improved the positioning accuracy of the stations, with the highest WRMS reduction being 2.719 mm and an average reduction of 1.03 mm. Removing these loads led to enhanced vertical positioning accuracy.

The impact of environmental loads on GNSS reference stations varied by region, with the greatest impact observed in the southwest region, followed by the northern China region and the northwestern frontier region. In contrast, the southeastern region experienced the least impact. Therefore, it is crucial to consider the influence of environmental loads on GNSS reference stations in regions outside of the southeast to improve their positioning accuracy. The primary reasons why removing environmental loads does not decrease the vertical amplitude of GNSS reference stations are twofold: first, the original GNSS time series may have weak periodicity or a weaker periodic pattern compared with the environmental load signals. Second, there can be a significant phase difference between the GNSS time series and the environmental load series, suggesting that the station's environmental loads are not strongly correlated with the GNSS vertical non-linear changes.

In its culmination, this study undertook an innovative exploration into the influence of SLMs on GNSS reference stations with different foundation types. After removing environmental loads, the annual period amplitude of the GNSS reference stations located on bedrock, 18 m soil layers, and 4–8 m soil layers decreased by 49.37%, 59.61%, and 46.48%, respectively. This finding indicates that GNSS reference stations situated on 18 m soil layers are more susceptible to environmental loads, potentially due to their concentration near fault zones and unstable geological environments.

In conclusion, this study provides valuable insights into the influence of environmental loads on GNSS reference stations in mainland China, contributing to improved positioning accuracy and a better understanding of the regional variability in environmental load effects on geodetic measurements.

**Author Contributions:** Conceptualization, Z.L.; Methodology, J.Z. and X.W.; Software, F.Y. and X.W.; Validation, J.Z. and Q.T.; Formal analysis, J.Z.; Investigation, J.Z., Z.L., F.Y. and X.L.; Resources, P.Z. and J.W.; Data curation, P.Z. and X.L.; Writing—original draft, J.Z., P.Z. and X.W.; Writing—review & editing, Z.L. and F.Y.; Supervision, Z.L.; Funding acquisition, J.W. All authors have read and agreed to the published version of the manuscript.

**Funding:** This research was funded by National Key Research and Development Program of China (2021YFC3000503; 2016YFB0501405).

**Data Availability Statement:** The datasets generated and analyzed during the current study are available from the corresponding author on reasonable request.

**Conflicts of Interest:** The authors declare no conflict of interest.

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
