# Peer review of "Assessing the Nonlinear Changes in Global Navigation Satellite System Vertical Time Series with Environmental Loading in Mainland China"

_remotesensing, doi:10.3390/rs15164115_

Round 1

Reviewer 1 Report

The subject matter addressed in the paper holds substantial significance, and it is likely to be of considerable interest to the readers. The method employed (filtering) demonstrates a moderate level of novelty, although there is room for enhancement. Nevertheless, its application on the network yields promising outcomes.   The conclusion of the paper, specifically pertaining to the investigation of GNSS reference stations situated on bedrock, soil layer depth, and their distinct behaviors under various influences, captured my particular attention.   To advance the paper's quality, I propose the following suggestions for potential improvement:   1. Provide clarity regarding the processing method utilized (my assumption is DD with Bern5.2), as well as explicit details regarding the concessions applied.   2. Consider conducting a comparative analysis between two types of solutions, namely DD and PPP.   3. Enhance data processing by incorporating a robust trend estimator for station velocities, such as MIDAS, which could amplify the certainty of the results while simultaneously minimizing errors.

Based on my assessment, I find the English proficiency exhibited in this paper to be of a high standard with only minor errors and potential recommendations for improvement from my perspective.

Page 4 – paragraph 1: by the following steps including - by the following steps, including

Page 4 – paragraph 3: used to quantifies – used to quantify

Page 4 – paragraph 4: andwtma – and wtma

Page 4 – paragraph 4: for example, is calculated - for example, which is calculated

Page 4 – last paragraph: Former studies emphasizes - Former studies emphasize

Page 6 – paragraph 1: The data presented in Table 1 is the statistics - The data presented in Table 1 are the statistics

Page 8 – last paragraph: number of stations experience - number of stations that experience

Reviewer 2 Report

Nonlinear Changes in Vertical Motion of GNSS is a hot topic in the community, the work write well, the authors do not highlight their innovations. Need to further refine the innovation points.

1.the title need to be simplified, e.g.: Assessing the Nonlinear Changes of GNSS vertical time series with Environmental Loading in Mainland China?

2. Missing necessary literature citations:e.g.:

[1]      Hu, S., Chen, K., Zhu, H., Xue, C., Wang, T., Yang, Z., & Zhao, Q. (2022). A comprehensive analysis of environmental loading effects on vertical GPS time series in yunnan, southwest China. Remote Sensing, 14(12), 2741.

[2]      He, X., Montillet, J. P., Hua, X., Yu, K., Jiang, W., & Zhou, F. (2017). Noise analysis for environmental loading effect on GPS position time series. Acta Geodynamica et Geomaterialia14(ARTICLE), 131-142.

[3]      Liu, B., Ma, X., Xing, X., Tan, J., Peng, W., & Zhang, L. (2022). Quantitative evaluation of environmental loading products and thermal expansion effect for correcting gnss vertical coordinate time series in taiwan. Remote Sensing, 14(18), 4480.

[4]      He, Y., Nie, G., Wu, S., & Li, H. (2022). Comparative analysis of the correction effect of different environmental loading products on global GNSS coordinate time series. Advances in Space Research, 70(11), 3594-3613.

3. AOH is not clear? Need more explain on it

4. Table 1 is hard to follow

5.for Figure 4: why there is large difference (from -4 to 8)? Same for Figure 5?

6. It is recommended that all the results of the same session in the article retain 2 decimal places

7. the presentation of Table 3 is rather strange

8. Figure 7-8 need provide high quality figures

9.the header of Table 4 also need simplified.

Reviewer 3 Report

This paper used 411 GNSS reference stations situated in mainland China to investigate the nonlinear changes in the vertical motion and assesses the influence of environmental load on vertical displacement. The researchers evaluate the effect of environmental load by calculating the change in annual cycle amplitude before and after its removal, focusing on its impact across regions with distinct foundation types. The results show that the environmental load leads to a considerable reduction of approximately 50.25% in the annual cycle amplitude of vertical motion for GNSS reference stations in mainland China. They also find that removing the environmental load results in a reduction of annual cycle amplitude by 49.37%, 59.61%, and 46.48%, respectively when categorizing GNSS reference stations based on their foundation type, namely bedrock, 18m soil layer, and 4-8m soil layer stations.

This paper considered the impact of environmental load that is crucial when analyzing the vertical motion of GNSS reference stations in mainland China, as it is essential for establishing a high-precision coordinate reference framework and studying the tectonic structure of the region.

The content of the paper is certainly of interest to remote sensing readers. However, it is some minor error need to be improved as blow.

Line 110: the presentation of CM and CF is not clear; some lines may be deleted. 

Line130: "Please mark the excluded sites in Figure 1.

Line184: Please provide a detailed explanation of the actual meaning of 'phase'

Line 175, 177,189,274: " the regions of China" may be considered to replace by "mainland of China";

Line 215:Missing annotation for 'abcd' in Figure 3

Line 289: " improvement underscores" may be replaced by "the significance of "

 Line 291: "However, there is one notable exception" maybe replaced by "Nevertheless, one notable exception is observed"

Line 440: "The reason behind this phenomenon may be attributed" may be replaced by  "This phenomenon may be attributed to"

Please explain why the use of 'WRMS' and 'amplitude variation' metrics.

Line 463: The influence of AOH on rock stations near the North-South fault zone has increased” may be replaced by “The influence of AOH on rock stations near the North-South fault zone has been found to be higher”

Round 2

Reviewer 2 Report

The author has improves the manuscript.

I only have one comment: AH should be abbreviation of several words, not a couple of description.

Need modify it before publication.
